# Uncertainty Sets for Image Classifiers using Conformal Prediction

**Anastasios N. Angelopoulos,**\* **Stephen Bates**\*, **Jitendra Malik, & Michael I. Jordan**
Departments of Electrical Engineering and Computer Sciences and Statistics
University of California, Berkeley
`{angelopoulos,stephenbates,malik,jordan}@cs.berkeley.edu`

## Abstract

Convolutional image classifiers can achieve high predictive accuracy, but quantifying their uncertainty remains an unresolved challenge, hindering their deployment in consequential settings. Existing uncertainty quantification techniques, such as Platt scaling, attempt to calibrate the network's probability estimates, but they do not have formal guarantees. We present an algorithm that modifies any classifier to output a *predictive set* containing the true label with a user-specified probability, such as 90%. The algorithm is simple and fast like Platt scaling, but provides a formal finite-sample coverage guarantee for every model and dataset. Our method modifies an existing conformal prediction algorithm to give more stable predictive sets by regularizing the small scores of unlikely classes after Platt scaling. In experiments on both Imagenet and Imagenet-V2 with ResNet-152 and other classifiers, our scheme outperforms existing approaches, achieving coverage with sets that are often factors of 5 to 10 smaller than a stand-alone Platt scaling baseline.

## 1 Introduction

Imagine you are a doctor making a high-stakes medical decision based on diagnostic information from a computer vision classifier. What would you want the classifier to output in order to make the best decision? This is not a casual hypothetical; such classifiers are already used in medical settings (e.g., Razzak et al., 2018; Lundervold & Lundervold, 2019; Li et al., 2014). A maximum-likelihood diagnosis with an accompanying probability may not be the most essential piece of information. To ensure the health of the patient, you must also rule in or rule out harmful diagnoses. In other words, even if the most likely diagnosis is a stomach ache, it is equally or more important to rule out stomach cancer. Therefore, you would want the classifier to give you—in addition to an estimate of the most likely outcome—actionable uncertainty quantification, such as a set of predictions that provably covers the true diagnosis with a high probability (e.g., 90%). This is called a *prediction set* (see Figure 1). Our paper describes a method for constructing prediction sets from any pre-trained image classifier that are formally guaranteed to contain the true class with the desired probability, relatively small, and practical to implement. Our method modifies a conformal predictor (Vovk et al., 2005) given in Romano et al. (2020) for the purpose of modern image classification in order to make it more stable in the presence of noisy small probability estimates. Just as importantly, we provide extensive evaluations and code for conformal prediction in computer vision.

Formally, for a discrete response $Y \in \mathcal{Y} = \{1, \ldots, K\}$ and a feature vector $X \in \mathbb{R}^d$, we desire an uncertainty set function, $\mathcal{C}(X)$, mapping a feature vector to a subset of $\{1, \ldots, K\}$ such that

$$P(Y \in \mathcal{C}(X)) \geq 1 - \alpha, \tag{1}$$

for a pre-specified confidence level $\alpha$ such as 10%. Conformal predictors like our method can modify any black-box classifier to output predictive sets that are rigorously guaranteed to satisfy the desired *coverage* property shown in Eq. (1). For evaluations, we focus on Imagenet classification

---

\*Equal contribution. Blog: `https://people.eecs.berkeley.edu/~angelopoulos/blog/posts/conformal-classification`

Figure 1: **Prediction set examples on Imagenet.** We show three examples of the class `fox squirrel` and the 95% prediction sets generated by RAPS to illustrate how the size of the set changes as a function of the difficulty of a test-time image.

using convolutional neural networks (CNNs) as the base classifiers, since this is a particularly challenging testbed. In this setting, $X$ would be the image and $Y$ would be the class label. Note that the guarantee in Eq. (1) is *marginal* over $X$ and $Y$—it holds on average, not for a particular image $X$.

A first approach toward this goal might be to assemble the set by including classes from highest to lowest probability (e.g., after Platt scaling and a softmax function; see Platt et al., 1999; Guo et al., 2017) until their sum just exceeds the threshold $1 - \alpha$. We call this strategy `naive` and formulate it precisely in Algorithm 1. There are two problems with `naive`: first, the probabilities output by CNNs are known to be incorrect (Nixon et al., 2019), so the sets from `naive` do not achieve coverage. Second, image classification models' tail probabilities are often badly miscalibrated, leading to large sets that do not faithfully articulate the uncertainty of the model; see Section 2.3. Moreover, smaller sets that achieve the same coverage level can be generated with other methods.

The coverage problem can be solved by picking a new threshold using holdout samples. For example, with $\alpha =$10%, if choosing sets that contain 93% estimated probability achieves 90% coverage on the holdout set, we use the 93% cutoff instead. We refer to this algorithm, introduced in Romano et al. (2020), as *Adaptive Prediction Sets* (APS). The APS procedure provides coverage but still produces large sets. To fix this, we introduce a regularization technique that tempers the influence of these noisy estimates, leading to smaller, more stable sets. We describe our proposed algorithm, *Regularized Adaptive Prediction Sets* (RAPS), in Algorithms 2 and 3 (with APS as a special case). As we will see in Section 2, both APS and RAPS are always guaranteed to satisfy Eq. (1)—regardless of model and dataset. Furthermore, we show that RAPS is guaranteed to have better performance than choosing a fixed-size set. Both methods impose negligible computational requirements in both training and evaluation, and output useful estimates of the model's uncertainty on a new image given, say, 1000 held-out examples.

In Section 3 we conduct the most extensive evaluation of conformal prediction in deep learning to date on Imagenet and Imagenet-V2. We find that RAPS sets always have smaller average size than `naive` and APSsets. For example, using a ResNeXt-101, `naive` does not achieve coverage, while APS and RAPS achieve it almost exactly. However, APS sets have an average size of 19, while RAPS sets have an average size of 2 at $\alpha = 10\%$ (Figure 2 and Table 1). We will provide an accompanying codebase that implements our method as a wrapper for any PyTorch classifier, along with code to exactly reproduce all of our experiments.

## 1.1 RELATED WORK

Reliably estimating predictive uncertainty for neural networks is an unsolved problem. Historically, the standard approach has been to train a Bayesian neural network to learn a distribution over network weights (Quinonero-Candela et al., 2005; MacKay, 1992; Neal, 2012; Kuleshov et al., 2018; Gal, 2016). This approach requires computational and algorithmic modifications; other approaches avoid these via ensembles (Lakshminarayanan et al., 2017; Jiang et al., 2018) or approximations of Bayesian inference (Riquelme et al., 2018; Sensoy et al., 2018). These methods also have major practical limitations; for example, ensembling requires training many copies of a neural network adversarially. Therefore, the most widely used strategy is ad-hoc *traditional calibration* of the softmax scores with Platt scaling (Platt et al., 1999; Guo et al., 2017; Nixon et al., 2019).

This work develops a method for uncertainty quantification based on *conformal prediction*. Originating in the online learning literature, conformal prediction is an approach for generating predictive sets that satisfy the coverage property in Eq. (1) (Vovk et al., 1999; 2005). We use a convenient data-splitting version known as *split conformal prediction* that enables conformal prediction meth-

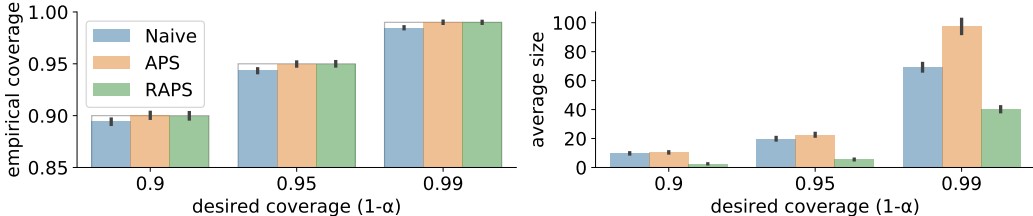

Figure 2: **Coverage and average set size on Imagenet for prediction sets from three methods.** All methods use a ResNet-152 as the base classifier, and results are reported for 100 random splits of Imagenet-Val, each of size 20K. See Section 3.1 for full details.

ods to be deployed for essentially any predictor (Papadopoulos et al., 2002; Lei et al., 2018). While mechanically very different from traditional calibration as discussed above, we will refer to our approach as *conformal calibration* to highlight that the two methodologies have overlapping but different goals.

Conformal prediction is a general framework, not a specific algorithm—important design decisions must be made to achieve the best performance for each context. To this end, Romano et al. (2020) and Cauchois et al. (2020) introduce techniques aimed at achieving coverage that is similar across regions of feature space, whereas Vovk et al. (2003); Hechtlinger et al. (2018) and Guan & Tibshirani (2019) introduce techniques aimed at achieving equal coverage for each class. While these methods have conceptual appeal, thus far there has been limited empirical evaluation of this general approach for state-of-the-art CNNs. Concretely, the only works that we are aware of that include some evaluation of conformal methods on ImageNet—the gold standard for benchmarking computer vision methods—are Hechtlinger et al. (2018), Park et al. (2019), Cauchois et al. (2020), and Messoudi et al. (2020), although in all four cases further experiments are needed to more fully evaluate their operating characteristics for practical deployment. At the heart of conformal prediction is the *conformal score* - a measure of similarity between labeled examples which is used to compare a new point to among those in a hold out set. Our theoretical contribution can be summarized as a modification of the conformal score from Romano et al. (2020) to have smaller, more stable sets. Lastly, there are alternative approaches to returning prediction sets not based on conformal prediction (Pearce et al., 2018; Zhang et al., 2018). These methods can be used as input to a conformal procedure to potentially improve performance, but they do not have finite-sample coverage guarantees when used alone.

## 2 METHODS

In developing uncertainty set methods to improve upon `naive`, we are guided by three desiderata. First and most importantly, the ***coverage desideratum*** says the sets must provide $1 - \alpha$ coverage, as discussed above. Secondly, the ***size desideratum*** says we want sets of small size, since these convey more detailed information and may be more useful in practice. Lastly, the ***adaptiveness desideratum*** says we want the sets to communicate instance-wise uncertainty: they should be smaller for easy test-time examples than for hard ones; see Figure 1 for an illustration. Coverage and size are obviously competing objectives, but size and adaptiveness are also often in tension. The size desideratum seeks small sets, while the adaptiveness desideratum seeks larger sets when the classi-

---

**Algorithm 1** Naive Prediction Sets

**Input:** $\alpha$, sorted scores $s$, associated permutation of classes $I$, boolean $rand$
1: **procedure** NAIVE($\alpha, s, I, rand$)
2:     $L \leftarrow 1$
3:     **while** $\sum_{i=1}^{L} s_i < 1 - \alpha$ **do**                            ▷ Stop if $1 - \alpha$ probability exceeded
4:         $L \leftarrow L + 1$
5:     **if** $rand$ **then**                                       ▷ Break ties randomly (explained in Appendix B)
6:         $U \leftarrow \text{Unif}(0, 1)$
7:         $V \leftarrow (\sum_{i=1}^{L} s_i - (1 - \alpha))/s_L$
8:         **if** $U \leq V$ **then**
9:             $L \leftarrow L - 1$
10:     **return** $\{I_1, ..., I_L\}$

**Output:** The $1 - \alpha$ prediction set, $\{I_1, ..., I_L\}$

---

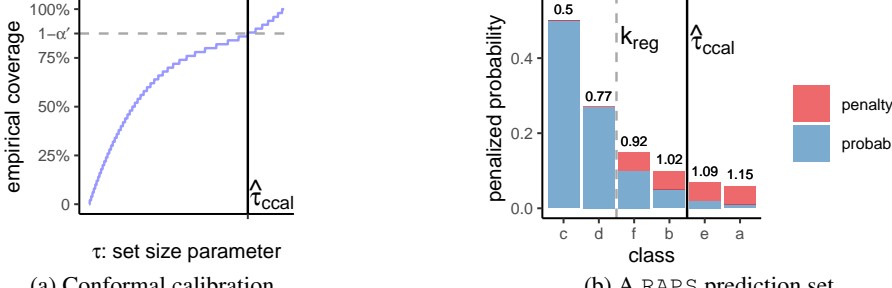

(a) Conformal calibration                    (b) A RAPS prediction set

Figure 3: **Visualizations of conformal calibration and RAPS sets.** In the left panel, the y-axis shows the empirical coverage on the conformal calibration set, and $1 - \alpha' = \lceil (n+1)(1-\alpha) \rceil / n$. In the right panel, the printed numbers indicate the cumulative probability plus penalty mass. For the indicated value $\hat{\tau}_{\mathrm{ccal}}$, the RAPS prediction set is {c, d, f, b}.

fier is uncertain. For example, always predicting a set of size five could achieve coverage, but it is not adaptive. As noted above, both APS and RAPS achieve correct coverage, and we will show that RAPS improves upon APS according to the other two desiderata.

We now turn to the specifics of our proposed method. We begin in Subsection 2.1 by describing an abstract data-splitting procedure called *conformal calibration* that enables the near-automatic construction of valid predictive sets (that is, sets satisfying Eq. (1)). Subsequently, in Subsection 2.2, we provide a detailed presentation of our procedure, with commentary in Section 2.3. In Subsection 2.4 we discuss the optimality of our procedure, proving that it is at least as good as the procedure that returns sets of a fixed size, unlike alternative approaches.

## 2.1 CONFORMAL CALIBRATION

We first review a general technique for producing valid prediction sets, following the articulation in Gupta et al. (2019). Consider a procedure that outputs a predictive set for each observation, and further suppose that this procedure has a tuning parameter $\tau$ that controls the size of the sets. (In RAPS, $\tau$ is the cumulative sum of the sorted, penalized classifier scores.) We take a small independent *conformal calibration set* of data, and then choose the tuning parameter $\tau$ such that the predictive sets are large enough to achieve $1 - \alpha$ coverage on this set. See Figure 3 for an illustration. This calibration step yields a choice of $\tau$, and the resulting set is formally guaranteed to have coverage $1 - \alpha$ on a future test point from the same distribution; see Theorem 1 below.

Formally, let $(X_i, Y_i)_{i=1,\dots,n}$ be an independent and identically distributed (i.i.d.) set of variables that was not used for model training. Further, let $\mathcal{C}(x, u, \tau) : \mathbb{R}^d \times [0,1] \times \mathbb{R} \to 2^{\mathcal{Y}}$ be a set-valued function that takes a feature vector $x$ to a subset of the possible labels. The second argument $u$ is included to allow for randomized procedures; let $U_1, \dots, U_n$ be i.i.d. uniform $[0,1]$ random variables that will serve as the second argument for each data point. Suppose that the sets are indexed by $\tau$ such that they are *nested*, meaning larger values of $\tau$ lead to larger sets:

$$\mathcal{C}(x, u, \tau_1) \subseteq \mathcal{C}(x, u, \tau_2) \quad \text{if} \quad \tau_1 \leq \tau_2. \tag{2}$$

To find a function that will achieve $1 - \alpha$ coverage on test data, we select the smallest $\tau$ that gives at least $1 - \alpha$ coverage on the conformal calibration set, with a slight correction to account for the finite sample size:

$$\hat{\tau}_{\mathrm{ccal}} = \inf \left\{ \tau : \frac{|\{i : Y_i \in \mathcal{C}(X_i, U_i, \tau)\}|}{n} \geq \frac{\lceil (n+1)(1-\alpha) \rceil}{n} \right\}. \tag{3}$$

The set function $\mathcal{C}(x, u, \tau)$ with this data-driven choice of $\tau$ is guaranteed to have correct finite-sample coverage on a fresh test observation, as stated formally next.

**Theorem 1** (Conformal calibration coverage guarantee). *Suppose* $(X_i, Y_i, U_i)_{i=1,\dots,n}$ *and* $(X_{n+1}, Y_{n+1}, U_{n+1})$ *are i.i.d. and let* $\mathcal{C}(x, u, \tau)$ *be a set-valued function satisfying the nesting property in Eq. (2). Suppose further that the sets* $\mathcal{C}(x, u, \tau)$ *grow to include all labels for large enough* $\tau$*: for all* $x \in \mathbb{R}^d$*,* $\mathcal{C}(x, u, \tau) = \mathcal{Y}$ *for some* $\tau$*. Then for* $\hat{\tau}_{\mathrm{ccal}}$ *defined as in Eq. (3), we have the following coverage guarantee:*

$$P\left(Y_{n+1} \in \mathcal{C}(X_{n+1}, U_{n+1}, \hat{\tau}_{\mathrm{ccal}})\right) \geq 1 - \alpha.$$

This is the same coverage property as Eq. (1) in the introduction, written in a more explicit manner. The result is not new—a special case of this result leveraging sample-splitting first appears in the regression setting in Papadopoulos et al. (2002), and the core idea of conformal prediction was introduced even earlier; see (Vovk et al., 2005).

As a technical remark, the theorem also holds if the observations to satisfy the weaker condition of exchangeability; see Vovk et al. (2005). In addition, for most families of set-valued functions $\mathcal{C}(x, u, \tau)$ there is a matching upper bound:

$$P\Big(Y_{n+1} \in \mathcal{C}(X_{n+1}, U_{n+1}, \hat{\tau}_{\text{ccal}})\Big) \leq 1 - \alpha + \frac{1}{n+1}.$$

Roughly speaking, this will hold whenever the sets grow smoothly in $\tau$. See Lei et al. (2018) for a formal statement of the required conditions.

## 2.2 OUR METHOD

Conformal calibration is a powerful general idea, allowing one to achieve the coverage desideratum for any choice of sets $\mathcal{C}(x, u, \tau)$. Nonetheless, this is not yet a full solution, since the quality of the resulting prediction sets can vary dramatically depending on the design of $\mathcal{C}(x, u, \tau)$. In particular, we recall the size and adaptiveness desiderata from Section 1—we want our uncertainty sets to be as small as possible while faithfully articulating the instance-wise uncertainty of each test point. In this section, we explicitly give our algorithm, which can be viewed as a special case of conformal calibration with the uncertainty sets $\mathcal{C}$ designed to extract information from CNNs.

Our algorithm has three main ingredients. First, for a feature vector $x$, the base model computes class probabilities $\hat{\pi}_x \in \mathbb{R}^k$, and we order the classes from most probable to least probable. Then, we add a regularization term to promote small predictive sets. Finally, we conformally calibrate the penalized prediction sets to guarantee coverage on future test points.

Formally, let $\rho_x(y) = \sum_{y'=1}^K \hat{\pi}_x(y') \mathbb{I}_{\{\hat{\pi}_x(y') > \hat{\pi}_x(y)\}}$ be the total probability mass of the set of labels that are more likely than $y$. These are all the labels that will be included before $y$ is included. In addition, let $o_x(y) = |\{y' \in \mathcal{Y} : \hat{\pi}_x(y') \geq \hat{\pi}_x(y)\}|$ be the ranking of $y$ among the label based on the probabilities $\hat{\pi}$. For example, if $y$ is the third most likely label, then $o_x(y) = 3$.[1] We take

$$\mathcal{C}^*(x, u, \tau) := \Big\{y \ : \ \rho_x(y) + \hat{\pi}_x(y) \cdot u + \underbrace{\lambda \cdot (o_x(y) - k_{reg})^+}_{\text{regularization}} \leq \tau \Big\}, \tag{4}$$

where $(z)^+$ denotes the positive part of $z$ and $\lambda, k_{reg} \geq 0$ are regularization hyperparameters that are introduced to encourage small set sizes. See Figure 3 for a visualization of a `RAPS` predictive set and Appendix E for a discussion of how to select $k_{reg}$ and $\lambda$.

Since this is the heart of our proposal, we carefully parse each term. First, the $\rho_x(y)$ term increases as $y$ ranges from the most probable to least probable label, so our sets will prefer to include the $y$ that are predicted to be the most probable. The second term, $\hat{\pi}_x(y) \cdot u$, is a randomized term to handle the fact that the value will jump discretely with the inclusion of each new $y$. The randomization term can never impact more than one value of $y$: there is at most one value of $y$ such that $y \in \mathcal{C}(x, 0, \tau)$ but $y \notin \mathcal{C}(x, 1, \tau)$. These first two terms can be viewed as the CDF transform after arranging the classes from most likely to least likely, randomized in the usual way to result in a continuous uniform random variable (cf. Romano et al., 2020). We discuss randomization further in Appendix B.

Lastly, the regularization promotes small set sizes: for values of $y$ that occur farther down the ordered list of classes, the term $\lambda \cdot (o_x(y) - k_{reg})^+$ makes that value of $y$ require a higher value of $\tau$ before it is included in the predictive set. For example, if $k_{reg} = 5$, then the sixth most likely value of $y$ has an extra penalty of size $\lambda$, so it will never be included until $\tau$ exceeds $\rho_x(y) + \hat{\pi}_x(y) \cdot u + \lambda$, whereas it enters when $\tau$ exceeds $\rho_x(y) + \hat{\pi}_x(y) \cdot u$ in the nonregularized version. Our method has the following coverage property:

**Proposition 1** (`RAPS` coverage guarantee). *Suppose* $(X_i, Y_i, U_i)_{i=1,\dots,n}$ *and* $(X_{n+1}, Y_{n+1}, U_{n+1})$ *are i.i.d. and let* $\mathcal{C}^*(x, u, \tau)$ *be defined as in Eq. (4). Suppose further that* $\hat{\pi}_x(y) > 0$ *for all* $x$ *and* $y$*. Then for* $\hat{\tau}_{\text{ccal}}$ *defined as in Eq. (3), we have the following coverage guarantee:*

$$1 - \alpha \leq P\Big(Y_{n+1} \in \mathcal{C}^*(X_{n+1}, U_{n+1}, \hat{\tau}_{\text{ccal}})\Big) \leq 1 - \alpha + \frac{1}{n+1}.$$

---

[1]For ease of notation, we assume distinct probabilities. Else, label-ordering ties should be broken randomly.

---

**Algorithm 2** RAPS Conformal Calibration

---

**Input:** $\alpha$; $s \in [0,1]^{n \times K}$, $I \in \{1, ..., K\}^{n \times K}$, and one-hot $y \in \{0,1\}^K$ corresponding respectively to the sorted scores, the associated permutation of indexes, and labels for each of $n$ examples in the calibration set; $k_{reg}$; $\lambda$; boolean $rand$

1: **procedure** RAPSC($\alpha$,$s$,$I$,$y$,$\lambda$)
2:     **for** $i \in \{1, \cdots, n\}$ **do**
3:         $L_i \leftarrow \{\ j\ :\ I_{i,j} = y_i\ \}$
4:         $E_i \leftarrow \Sigma_{j=0}^{L_i} s_{i,j} + \lambda (L_i - k_{reg} + 1)^+$
5:         **if** $rand$ **then**
6:             $U \sim \text{Unif}(0,1)$
7:             $E_i \leftarrow E_i - s_{i,L_i} + U * s_{i,L_i}$
8:     $\hat{\tau}_{ccal} \leftarrow$ the $\lceil (1-\alpha)(1+n) \rceil$ largest value in $\{E_i\}_{i=1}^n$
9:     **return** $\hat{\tau}_{ccal}$

**Output:** The generalized quantile, $\hat{\tau}_{ccal}$                   $\triangleright$ The value in Eq. (3)

---

**Algorithm 3** RAPS Prediction Sets

---

**Input:** $\alpha$, sorted scores $s$ and the associated permutation of classes $I$ for a test-time example, $\hat{\tau}_{ccal}$ from Algorithm 2, $k_{reg}$, $\lambda$, boolean $rand$

1: **procedure** RAPS($\alpha, s, I, \hat{\tau}_{ccal}, k_{reg}, \lambda, rand$)
2:     $L \leftarrow |\ j \in \mathcal{Y}\ :\ \Sigma_{i=0}^{j} s_i + \lambda (L - k_{reg})^+ \leq \hat{\tau}_{ccal}\ | + 1$
3:     $V \leftarrow (\hat{\tau}_{ccal} - \Sigma_{i=0}^{L-1} s_i - \lambda (L - k_{reg})^+ + s_{L-1})/s_{L-1}$
4:     **if** $rand$ & $V \leq U \sim \text{Unif}(0,1)$ **then**
5:         $L \leftarrow L - 1$
6:     **return** $\mathcal{C} = \{I_1, ... I_L\}$                 $\triangleright$ The $L$ most likely classes

**Output:** The $1 - \alpha$ confidence set, $\mathcal{C}$               $\triangleright$ The set in Eq. (4)

---

Note that the first inequality is a corollary of Theorem 1, and the second inequality is a special case of the remark in Section 2.1. The restriction that $\hat{\pi}_x(y) > 0$ is not necessary for the first inequality.

### 2.3 WHY REGULARIZE?

In our experiments, the sets from APS are larger than necessary, because APS is sensitive to the noisy probability estimates far down the list of classes. This noise leads to a *permutation problem* of unlikely classes, where ordering of the classes with small probability estimates is determined mostly by random chance. If 5% of the true classes from the calibration set are deep in the tail due to the permutation problem, APS will choose large 95% predictive sets; see Figure 2. The inclusion of the RAPS regularization causes the algorithm to avoid using the unreliable probabilities in the tail; see Figure 4. We discuss how RAPS improves the adaptiveness of APS in Section 4 and Appendix E.

### 2.4 OPTIMALITY CONSIDERATIONS

To complement these experimental results, we now formally prove that RAPS with the correct regularization parameters will always dominate the simple procedure that returns a fixed set size. (Section 3.5 shows the parameters are easy to select and RAPS is not sensitive to their values). For a feature vector $x$, let $\hat{y}_{(j)}(x)$ be the label with the $j$th highest predicted probability. We define the *top-k predictive sets* to be $\{\hat{y}_{(1)}(x), \ldots, \hat{y}_{(k)}(x)\}$.

**Proposition 2** (RAPS dominates top-k sets)**.** *Suppose* $(X_i, Y_i, U_i)_{i=1,...,n}$ *and* $(X_{n+1}, Y_{n+1}, U_{n+1})$ *are i.i.d. draws. Let* $k^*$ *be the smallest* $k$ *such that the top-k predictive sets have coverage at least* $\lceil (n+1)(1-\alpha) \rceil / n$ *on the conformal calibration points* $(X_i, Y_i)_{i=1,...,n}$*. Take* $\mathcal{C}^*(x, u, \tau)$ *as in Eq. (4) with any* $k_{reg} \leq k^*$ *and* $\lambda = 1$*. Then with* $\hat{\tau}_{\text{ccal}}$ *chosen as in Eq. (3), we have*

$$\mathcal{C}^*(X_{n+1}, U_{n+1}, \hat{\tau}_{\text{ccal}}) \ \subseteq \ \{\hat{y}_{(1)}(x), \ldots, \hat{y}_{(k^*)}(x)\}.$$

In words, the RAPS procedure with heavy regularization will be at least as good as the top-$k$ procedure in the sense that it has smaller or same average set size while maintaining the desired coverage level. This is not true of either the naive baseline or the APS procedure; Table 2 shows that these two procedures usually return predictive sets with size much larger than $k^*$.

## 3 EXPERIMENTS

| Model | Accuracy | | Coverage | | | | Size | | | |
|---|---|---|---|---|---|---|---|---|---|---|
| | Top-1 | Top-5 | Top K | Naive | APS | RAPS | Top K | Naive | APS | RAPS |
| ResNeXt101 | 0.793 | 0.945 | 0.900 | 0.889 | 0.900 | 0.900 | 2.42 | 17.1 | 19.7 | **2.00** |
| ResNet152 | 0.783 | 0.94 | 0.900 | 0.895 | 0.900 | 0.900 | 2.63 | 9.78 | 10.4 | **2.11** |
| ResNet101 | 0.774 | 0.936 | 0.900 | 0.896 | 0.900 | 0.900 | 2.83 | 10.3 | 10.7 | **2.25** |
| ResNet50 | 0.761 | 0.929 | 0.900 | 0.896 | 0.900 | 0.900 | 3.14 | 11.8 | 12.3 | **2.57** |
| ResNet18 | 0.698 | 0.891 | 0.900 | 0.895 | 0.900 | 0.900 | 5.72 | 15.5 | 16.2 | **4.43** |
| DenseNet161 | 0.771 | 0.936 | 0.900 | 0.894 | 0.900 | 0.900 | 2.84 | 11.2 | 12.1 | **2.29** |
| VGG16 | 0.716 | 0.904 | 0.900 | 0.895 | 0.901 | 0.900 | 4.75 | 13.4 | 14.1 | **3.54** |
| Inception | 0.695 | 0.887 | 0.900 | 0.885 | 0.900 | 0.901 | 6.30 | 75.4 | 89.1 | **5.32** |
| ShuffleNet | 0.694 | 0.883 | 0.900 | 0.891 | 0.900 | 0.900 | 6.46 | 28.9 | 31.9 | **5.05** |

Table 1: **Results on Imagenet-Val.** We report coverage and size of the optimal, randomized fixed sets, `naive`, `APS`, and `RAPS` sets for nine different Imagenet classifiers. The median-of-means for each column is reported over 100 different trials. See Section 3.1 for full details.

In this section we report on experiments that study the performance of the predictive sets from `naive`, `APS`, and `RAPS`, evaluating each based on the three desiderata above. We begin with a brief preview of the experiments. In **Experiment 1**, we evaluate `naive`, `APS`, and `RAPS` on Imagenet-Val. Both `APS` and `RAPS` provided almost exact coverage, while `naive` sets had coverage slightly below the specified level. `APS` has larger sets on average than `naive` and `RAPS`. `RAPS` has a much smaller average set size than `APS` and `naive`. In **Experiment 2**, we repeat Experiment 1 on Imagenet-V2, and the conclusions still hold. In **Experiment 3**, we produce histograms of set sizes for `naive`, `APS`, and `RAPS` for several different values of $\lambda$, illustrating a simple tradeoff between set size and adaptiveness. In **Experiment 4**, we compute histograms of `RAPS` sets stratified by image difficulty, showing that `RAPS` sets are smaller for easier images than for difficult ones. In **Experiment 5**, we report the performance of `RAPS` with many values of the tuning parameters.

In our experiments, we use nine standard, pretrained Imagenet classifiers from the `torchvision` repository (Paszke et al., 2019) with standard normalization, resize, and crop parameters. Before applying `naive`, `APS`, or `RAPS`, we calibrated the classifiers using the standard temperature scaling/Platt scaling procedure as in Guo et al. (2017) on the calibration set. Thereafter, `naive`, `APS`, and `RAPS` were applied, with `RAPS` using a data-driven choice of parameters described in Appendix E. We use the randomized versions of these algorithms—see Appendix B for a discussion.

### 3.1 EXPERIMENT 1: COVERAGE VS SET SIZE ON IMAGENET

In this experiment, we calculated the coverage and mean set size of each procedure for two different choices of $\alpha$. Over 100 trials, we randomly sampled two subsets of Imagenet-Val: one conformal calibration subset of size 20K and one evaluation subset of size 20K. The median-of-means over trials for both coverage and set size are reported in Table 1. Figure 2 illustrates the performances of `naive`, `APS`, and `RAPS`; `RAPS` has much smaller sets than both `naive` and `APS`, while achieving coverage. We also report results from a conformalized fixed-k procedure, which finds the smallest fixed set size achieving coverage on the holdout set, $k^*$, then predicts sets of size $k^* - 1$ or $k^*$ on new examples in order to achieve exact coverage; see Algorithm 4 in Appendix E.

### 3.2 EXPERIMENT 2: COVERAGE VS SET SIZE ON IMAGENET-V2

The same procedure as Experiment 1 was repeated on Imagenet-V2, with exactly the same normalization, resize, and crop parameters. The size of the calibration and evaluation sets was 5K, since Imagenet-V2 is a smaller dataset. The result shows that our method can still provide coverage even for models trained on different distributions, as long as the conformal calibration set comes from the new distribution. The variance of the coverage is higher due to having less data.

### 3.3 EXPERIMENT 3: SET SIZES OF NAIVE, APS, AND RAPS ON IMAGENET

We investigate the effect of regularization in more detail. For three values of $\lambda$, we collected the set sizes produced by each of `naive`, `APS`, and `RAPS` and report their histograms in Figure 4.

### 3.4 EXPERIMENT 4: ADAPTIVENESS OF RAPS ON IMAGENET

We now show that `RAPS` sets are smaller for easy images than hard ones, addressing the adaptiveness desideratum. Table 4 reports the size-stratified coverages of `RAPS` at the 90% level with $k_{reg} = 5$ and different choices of $\lambda$. When $\lambda$ is small, `RAPS` allows sets to be large. But when $\lambda = 1$, `RAPS`

| | Accuracy | | Coverage | | | | Size | | | |
|---|---|---|---|---|---|---|---|---|---|---|
| Model | Top-1 | Top-5 | Top K | Naive | APS | RAPS | Top K | Naive | APS | RAPS |
| ResNeXt101 | 0.678 | 0.874 | 0.900 | 0.888 | 0.899 | 0.899 | 7.48 | 43.0 | 50.8 | **6.18** |
| ResNet152 | 0.67 | 0.876 | 0.899 | 0.896 | 0.900 | 0.900 | 7.18 | 25.8 | 27.2 | **5.69** |
| ResNet101 | 0.657 | 0.859 | 0.901 | 0.894 | 0.900 | 0.898 | 9.21 | 28.7 | 30.7 | **6.93** |
| ResNet50 | 0.634 | 0.847 | 0.898 | 0.894 | 0.899 | 0.900 | 10.3 | 30.3 | 32.3 | **7.80** |
| ResNet18 | 0.572 | 0.802 | 0.902 | 0.895 | 0.900 | 0.900 | 17.5 | 35.3 | 37.4 | **13.3** |
| DenseNet161 | 0.653 | 0.862 | 0.902 | 0.895 | 0.901 | 0.901 | 8.6 | 29.9 | 32.4 | **6.93** |
| VGG16 | 0.588 | 0.817 | 0.902 | 0.897 | 0.900 | 0.899 | 15.1 | 31.9 | 32.8 | **11.2** |
| Inception | 0.573 | 0.797 | 0.900 | 0.893 | 0.900 | 0.899 | 21.8 | 145.0 | 155.0 | **20.5** |
| ShuffleNet | 0.559 | 0.781 | 0.899 | 0.892 | 0.900 | 0.899 | 26.0 | 66.2 | 71.7 | **22.5** |

Table 2: **Results on Imagenet-V2.** We report coverage and size of the optimal, randomized fixed sets, `naive`, `APS`, and `RAPS` sets for nine different Imagenet classifiers. The median-of-means for each column is reported over 100 different trials at the 10% level. See Section 3.2 for full details.

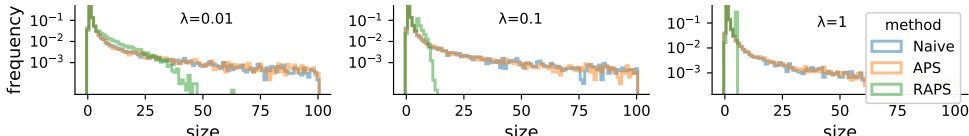

Figure 4: **Set sizes produced with ResNet-152.** See Section 3.3 for details.

clips sets to be a maximum of size 5. Table 7 (in the Appendix) stratifies by image difficulty, showing that `RAPS` sets are small for easy examples and large for hard ones. Experiments 3 and 4 together illustrate the tradeoff between adaptiveness and size: as the average set size decreases, the `RAPS` procedure truncates sets larger than the smallest fixed set that provides coverage, taming the heavy tail of the `APS` procedure. Since `RAPS` with large $\lambda$ undercovers hard examples, it must compensate by taking larger sets for easy examples to ensure the $1 - \alpha$ marginal coverage guarantee. However, the size only increases slightly since easy images are more common than hard ones, and the total probability mass can often exceed $\hat{\tau}_{ccal}$ by including only one more class. If this behavior is not desired, we can instead automatically pick $\lambda$ to optimize the adaptiveness of `RAPS`; see Section 4.

### 3.5 EXPERIMENT 5: CHOICE OF TUNING PARAMETERS

While any value of the tuning parameters $\lambda$ and $k_{reg}$ lead to coverage (Proposition 1), some values will lead to smaller sets. In Experiments 1 and 2, we chose $k_{reg}$ and $\lambda$ adaptively from data (see Appendix E), achieving strong results for all models and choices of the coverage level. Table 3 gives the performance of `RAPS` with many choices of $k_{reg}$ and $\lambda$ for ResNet-152.

## 4 ADAPTIVENESS AND CONDITIONAL COVERAGE

In this section, we point to a definition of adaptiveness that is more natural for the image classification setting than the existing notion of conditional coverage. We show that `APS` does not satisfy conditional coverage, and that `RAPS` with small $\lambda$ outperforms it in terms of adaptiveness.

We say that a set-valued predictor $\mathcal{C} : \mathbb{R}^d \to 2^{\mathcal{Y}}$ satisfies exact *conditional coverage* if $P(Y \in \mathcal{C}(X) \mid X = x) = 1 - \alpha$ for each $x$. Distribution-free guarantees on conditional coverage are impossible (Vovk, 2012; Lei & Wasserman, 2014), but many algorithms try to satisfy it approximately (Romano et al., 2019; 2020; Cauchois et al., 2020). In a similar spirit, Tibshirani et al. (2019) suggest a notion of local conditional coverage, where one asks for coverage in a neighborhood of each point, weighted according to a chosen kernel. Cauchois et al. (2020) introduce the *worst-case slab* metric for measuring violations of the conditional coverage property. We present a different way of measuring violations of conditional coverage.

**Proposition 3.** *Suppose $P(Y \in \mathcal{C}(X) \mid X = x) = 1 - \alpha$ for each $x \in \mathbb{R}^d$. Then,*
$P(Y \in \mathcal{C}(X) \mid \{|C(X)| \in \mathcal{A}\}) = 1 - \alpha$ *for any $\mathcal{A} \subset \{0, 1, 2, \dots\}$.*

In words, if conditional coverage holds, then coverage holds after stratifying by set size. Based on this result, In Appendix E, we introduce the *size-stratified coverage violation* criterion, a simple and pragmatic way of quantifying adaptiveness. Then, we automatically tune $\lambda$ on this metric so `RAPS` markedly outperforms the adaptiveness of `APS` (see Table 8).

| $k_{reg}|\lambda$ | 0 | 1e-4 | 1e-3 | 0.01 | 0.02 | 0.05 | 0.2 | 0.5 | 0.7 | 1.0 |
|---|---|---|---|---|---|---|---|---|---|---|
| 1 | 11.2 | 10.2 | 7.0 | 3.6 | 2.9 | 2.3 | 2.1 | 2.3 | 2.2 | 2.2 |
| 2 | 11.2 | 10.2 | 7.1 | 3.7 | 3.0 | 2.4 | 2.1 | 2.3 | 2.2 | 2.2 |
| 5 | 11.2 | 10.2 | 7.2 | 3.9 | 3.4 | 2.9 | 2.6 | 2.5 | 2.5 | 2.5 |
| 10 | 11.2 | 10.2 | 7.4 | 4.5 | 4.0 | 3.6 | 3.4 | 3.4 | 3.4 | 3.4 |
| 50 | 11.2 | 10.6 | 8.7 | 7.2 | 7.0 | 6.9 | 6.9 | 6.9 | 6.9 | 6.9 |

Table 3: Set sizes of `RAPS` with parameters $k_{reg}$ and $\lambda$, a ResNet-152, and coverage level 90%.

| size | $\lambda = 0$ | | $\lambda = 0.001$ | | $\lambda = 0.01$ | | $\lambda = 0.1$ | | $\lambda = 1$ | |
|---|---|---|---|---|---|---|---|---|---|---|
| | cnt | cvg | cnt | cvg | cnt | cvg | cnt | cvg | cnt | cvg |
| 0 to 1 | 11627 | 0.88 | 11539 | 0.88 | 11225 | 0.89 | 10476 | 0.92 | 10027 | 0.93 |
| 2 to 3 | 3687 | 0.91 | 3702 | 0.91 | 3741 | 0.92 | 3845 | 0.93 | 3922 | 0.94 |
| 4 to 6 | 1239 | 0.91 | 1290 | 0.91 | 1706 | 0.92 | 4221 | 0.89 | 6051 | 0.83 |
| 7 to 10 | 688 | 0.93 | 765 | 0.93 | 1314 | 0.91 | 1436 | 0.71 | 0 | |
| 11 to 100 | 2207 | 0.94 | 2604 | 0.93 | 2014 | 0.86 | 22 | 0.59 | 0 | |
| 101 to 1000 | 552 | 0.97 | 100 | 0.90 | 0 | | 0 | | 0 | |

Table 4: **Coverage conditional on set size.** We report average coverage of images stratified by the size of the set output by `RAPS` using a ResNet-152 for varying $\lambda$. The marginal coverage rate is 90%.

In Table 4, we report on the coverage of `APS` and `RAPS`, stratified by the size of the prediction set. Turning our attention to the $\lambda = 0$ column, we see that when `APS` outputs a set of size $101 - 1000$, `APS` has coverage 97%, substantially higher than 90% nominal rate. By Proposition 3, we conclude that `APS` is not achieving exact conditional coverage, because the scores are far from the oracle probabilities. The `APS` procedure still achieves marginal coverage by overcovering hard examples and undercovering easy ones, an undesirable behavior. Alternatively, `RAPS` can be used to regularize the set sizes—for $\lambda = .001$ to $\lambda = .01$ the coverage stratified by set size is more balanced. In summary, even purely based on the adaptiveness desideratum, `RAPS` with light regularization is preferable to `APS`. Note that as the size of the training data increases, as long as $\hat{\pi}$ is consistent, `naive` and `APS` will become more stable, and so we expect less regularization will be needed.

Lastly, we argue that conditional coverage is a poor notion of adaptiveness when the best possible model (i.e., one fit on infinite data) has high accuracy. Given such a model, the oracle procedure from Romano et al. (2020) would return the correct label with probability $1 - \alpha$ and the empty set with probability $\alpha$. That is, having correct conditional coverage for high-signal problems where $Y$ is perfectly determined by $X$ requires a perfect classifier. In our experiments on ImageNet, `APS` does not approximate this behavior. Therefore, conditional coverage isn't the right goal for prediction sets with realistic sample sizes. Proposition 3 suggests a relaxation. We could require that we have the right coverage, no matter the size of the prediction set: $P(Y \in \mathcal{C}(X) \mid \{|C(x)| \in \mathcal{A}\}) \geq 1 - \alpha$ for any $\mathcal{A} \subset \{0, 1, 2, \dots\}$; Appendix E.2 develops this idea. We view this as a promising way to reason about adaptiveness in high-signal problems such as image classification.

## 5 DISCUSSION

For classification tasks with many possible labels, our method enables a researcher to take any base classifier and return predictive sets guaranteed to achieve a pre-specified error level, such as 90%, while retaining small average size. It is simple to deploy, so it is an attractive, automatic way to quantify the uncertainty of image classifiers—an essential task in such settings as medical diagnostics, self-driving vehicles, and flagging dangerous internet content. Predictive sets in computer vision (from `RAPS` and other conformal methods) have many further uses, since they systematically identify hard test-time examples. Finding such examples is useful in active learning where one only has resources to label a small number of points. In a different direction, one can improve efficiency of a classifier by using a cheap classifier outputting a prediction set first, and an expensive one only when the cheap classifier outputs a large set (a *cascade*; see, e.g., Li et al. 2015), and Fisch et al. (2021) for an implementation of conformal prediction in this setting. One can also use predictive sets during model development to identify failure cases and outliers and suggest strategies for improving its performance. Prediction sets are most useful for problems with many classes; returning to our initial medical motivation, we envision `RAPS` could be used by a doctor to automatically screen for a large number of diseases (e.g. via a blood sample) and refer the patient to relevant specialists.

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

# A PROOFS

*Theorem 1.* Let $s(x, u, y) = \inf_\tau \{y \in \mathcal{C}(x, u, \tau)\}$, and let $s_i = s(X_i, U_i, Y_i)$ for $i = 1, \ldots, n$. Then

$$\{y : s(x, u, y) \leq \tau\} = \{y : y \in \mathcal{C}(x, u, \tau)\}$$

because $\mathcal{C}(x, u, \tau)$ is a finite set growing in $\tau$ by the assumption in Eq. (2). Thus,

$$\{\tau : |\{i : s_i \leq \tau\}| \geq \lceil (1 - \alpha)(n + 1) \rceil\} = \left\{\tau : \frac{|\{i : Y_i \in \mathcal{C}(X_i, U_i, \tau)\}|}{n} \geq \frac{\lceil (n + 1)(1 - \alpha) \rceil}{n}\right\}.$$

Considering the left expression, the infimum over $\tau$ of the set on the left hand side is the $\lceil (1 - \alpha)(n + 1) \rceil$ smallest value of the $s_i$, so this is the value of $\hat{\tau}_{\text{ccal}}$. Since $s_1, \ldots, s_n, s(X_{n+1}, U_{n+1}, Y_{n+1})$ are exchangeable random variables, $|\{i : s(X_{n+1}, U_{n+1}, Y_{n+1}) > s_i\}|$ is stochastically dominated by the discrete uniform distribution on $\{0, 1, \ldots, n\}$. We thus have that

$$\begin{aligned}
P(Y_{n+1} \notin \mathcal{C}(X_{n+1}, U_{n+1}, \hat{\tau}_{\text{ccal}})) &= P(s(X_{n+1}, U_{n+1}, Y_{n+1}) > \hat{\tau}_{\text{ccal}}) \\
&= P(|\{i : s(X_{n+1}, U_{n+1}, Y_{n+1}) > s_i\}| \geq \lceil (n + 1)(1 - \alpha) \rceil) \\
&= P\left(\frac{|\{i : s(X_{n+1}, U_{n+1}, Y_{n+1}) > s_i\}|}{n + 1} \geq \frac{\lceil (n + 1)(1 - \alpha) \rceil}{n + 1}\right) \\
&\leq \alpha.
\end{aligned}$$

$\square$

*Proposition 1.* The lower bound follows from Theorem 1. To prove the upper bound, using the result from Theorem 2.2 of Lei et al. (2018) it suffices to show that the variables $s(X_i, U_i, Y_i) = \inf\{\tau : Y_i \in \mathcal{C}(X_i, U_i, \tau)\}$ are almost surely distinct. To this end, note that that

$$s(X_i, U_i, Y_i) = \rho_{X_i}(Y_i) + \hat{\pi}_{X_i}(Y_i) \cdot U_i + \lambda(o_{X_i}(Y_i) - k_{reg})^+,$$

and due to the middle term of the sum, these values are distinct almost surely provided $\hat{\pi}_{X_i}(Y_i) > 0$. $\square$

*Proposition 2.* We first show that $\hat{\tau}_{\text{ccal}} \leq 1 + k^* - k_{reg}$. Note that since at least $\lceil (1 - \alpha)(n + 1) \rceil$ of the conformal calibration points are covered by a set of size $k^*$, at least $\lceil (1 - \alpha)(n + 1) \rceil$ of the $E_i$ in Algorithm 2 are less than or equal to $1 + k^* - k_{reg}$. Thus, by the definition of $\hat{\tau}_{\text{ccal}}$, we have that it is less than or equal to $1 + k^* - k_{reg}$. Then, note that by the definition of $\mathcal{C}^*$ in Eq. (4), we have that

$$|\mathcal{C}^*(X_{n+1}, U_{n+1}, \hat{\tau}_{\text{ccal}})| \leq k^*.$$

as long as $\hat{\tau}_{\text{ccal}} \leq 1 + k^* - k_{reg}$, since for the $k^* + 1$ most likely class, the sum in Eq. (4) will exceed $\lambda \cdot (1 + k^* - k_{reg}) = (1 + k^* - k_{reg}) \geq \hat{\tau}_{\text{ccal}}$, and so the $k^* + 1$ class will not be in the set. $\square$

*Proposition 3.* Suppose $P(Y \in \mathcal{C}(X) \mid X = x) = 1 - \alpha$ for each $x \in \mathbb{R}^d$. Then,

$$\begin{aligned}
P(Y \in \mathcal{C}(X) \mid |C(X)| \in \mathcal{A}) &= \frac{\int_x P(Y \in \mathcal{C}(x) \mid X = x)\mathbb{I}_{\{|\mathcal{C}(x)| \in \mathcal{A}\}}dP(x)}{P(|\mathcal{C}(X)| \in \mathcal{A})} \\
&= \frac{\int_x (1 - \alpha)\mathbb{I}_{\{|\mathcal{C}(x)| \in \mathcal{A}\}}dP(x)}{P(|\mathcal{C}(X)| \in \mathcal{A})} \\
&= 1 - \alpha.
\end{aligned}$$

$\square$

| | Accuracy | | Coverage | | | | Size | | | |
|---|---|---|---|---|---|---|---|---|---|---|
| Model | Top-1 | Top-5 | Top K | Naive | APS | RAPS | Top K | Naive | APS | RAPS |
| ResNeXt101 | 0.794 | 0.945 | 0.905 | 0.938 | 0.950 | 0.950 | 5.64 | 36.4 | 46.3 | **4.21** |
| ResNet152 | 0.783 | 0.940 | 0.950 | 0.943 | 0.950 | 0.950 | 6.36 | 19.6 | 22.5 | **4.40** |
| ResNet101 | 0.774 | 0.936 | 0.950 | 0.944 | 0.950 | 0.950 | 6.79 | 20.6 | 23.2 | **4.79** |
| ResNet50 | 0.762 | 0.929 | 0.951 | 0.943 | 0.950 | 0.950 | 8.12 | 22.9 | 26.2 | **5.57** |
| ResNet18 | 0.698 | 0.891 | 0.950 | 0.943 | 0.950 | 0.950 | 16.0 | 28.9 | 33.2 | **11.7** |
| DenseNet161 | 0.772 | 0.936 | 0.950 | 0.942 | 0.950 | 0.950 | 6.84 | 23.4 | 28.0 | **5.09** |
| VGG16 | 0.716 | 0.904 | 0.950 | 0.943 | 0.950 | 0.950 | 12.9 | 24.6 | 27.8 | **8.98** |
| Inception | 0.695 | 0.887 | 0.950 | 0.937 | 0.950 | 0.950 | 20.3 | 142.0 | 168.0 | **18.5** |
| ShuffleNet | 0.694 | 0.883 | 0.950 | 0.940 | 0.950 | 0.950 | 19.3 | 58.7 | 71.6 | **16.3** |

Table 5: **Results on Imagenet-Val.** We report coverage and size of the optimal, randomized fixed sets, `naive`, `APS`, and `RAPS` sets for nine different Imagenet classifiers. The median-of-means for each column is reported over 100 different trials. See Section 3.1 for full details.

| | Accuracy | | Coverage | | | | Size | | | |
|---|---|---|---|---|---|---|---|---|---|---|
| Model | Top-1 | Top-5 | Top K | Naive | APS | RAPS | Top K | Naive | APS | RAPS |
| ResNeXt101 | 0.678 | 0.875 | 0.950 | 0.937 | 0.950 | 0.950 | 21.7 | 86.2 | 107.0 | **18.5** |
| ResNet152 | 0.670 | 0.876 | 0.951 | 0.944 | 0.950 | 0.950 | 21.3 | 51.0 | 56.6 | **16.2** |
| ResNet101 | 0.656 | 0.86 | 0.95 | 0.944 | 0.950 | 0.949 | 25.7 | 55.8 | 63.1 | **19.1** |
| ResNet50 | 0.634 | 0.847 | 0.949 | 0.944 | 0.949 | 0.950 | 29.5 | 58.6 | 65.9 | **21.5** |
| ResNet18 | 0.572 | 0.802 | 0.950 | 0.942 | 0.950 | 0.949 | 48.3 | 65.0 | 74.0 | **35.3** |
| DenseNet161 | 0.653 | 0.861 | 0.951 | 0.941 | 0.950 | 0.949 | 25.9 | 60.0 | 72.7 | **20.4** |
| VGG16 | 0.588 | 0.816 | 0.950 | 0.943 | 0.950 | 0.949 | 38.5 | 57.8 | 63.9 | **26.4** |
| Inception | 0.573 | 0.797 | 0.950 | 0.943 | 0.949 | 0.950 | 73.1 | 253.0 | 275.0 | **70.2** |
| ShuffleNet | 0.560 | 0.781 | 0.950 | 0.941 | 0.949 | 0.949 | 80.0 | 125.0 | 140.0 | **67.4** |

Table 6: **Results on Imagenet-V2.** We report coverage and size of the optimal, randomized fixed sets, `naive`, `APS`, and `RAPS` sets for nine different Imagenet classifiers. The median-of-means for each column is reported over 100 different trials at the 5% level. See Section 3.2 for full details.

## B    RANDOMIZED PREDICTORS

The reader may wonder why we choose to use a randomized procedure. The randomization is needed to achieve $1 - \alpha$ coverage exactly, which we will explain via an example. Note that the randomization is of little practical importance, since the predictive set output by the randomized procedure will differ from the that of the non-randomized procedure by at most one element.

Turning to an example, assume for a particular input image we expect a set of size $k$ to have 91% coverage, and a set of size $k - 1$ to have 89% coverage. In order to achieve our desired coverage of 90%, we randomly choose size $k$ or $k - 1$ with equal probability. In general, the probabilities will not be equal, but rather chosen so the weighted average of the two coverages is exactly 90%. If a user of our method desires deterministic sets, it is easy to turn off this randomization with a single flag, resulting in slightly conservative sets.

## C    IMAGENET AND IMAGENETV2 RESULTS FOR $\alpha = 5\%$

We repeated Experiments 1 and 2 with $\alpha = 5\%$. See the results in Tables 5 and 6.

## D    COVERAGE AND SIZE CONDITIONAL ON IMAGE DIFFICULTY

In order to probe the adaptiveness properties of `APS` and `RAPS` we stratified coverage and size by image difficulty (the position of the true label in the list of most likely to least likely classes, based on the classifier predictions) in Table 7. With increasing $\lambda$, coverage decreases for more difficult images and increases for easier ones. In the most difficult regime, even though `APS` can output large sets, those sets still rarely contain the true class. This suggests regularization is a sensible way to stabilize the sets. As a final word on Table 7, notice that as $\lambda$ increases, coverage improves for the more common medium-difficulty examples, although not for very rare and difficult ones.

| | | $\lambda = 0$ | | $\lambda = 0.001$ | | $\lambda = 0.01$ | | $\lambda = 0.1$ | | $\lambda = 1$ | |
|---|---|---|---|---|---|---|---|---|---|---|---|
| difficulty | count | cvg | sz | cvg | sz | cvg | sz | cvg | sz | cvg | sz |
| 1 | 15668 | 0.95 | 5.2 | 0.95 | 3.8 | 0.96 | 2.5 | 0.97 | 2.0 | 0.98 | 2.0 |
| 2 to 3 | 2578 | 0.78 | 15.7 | 0.78 | 10.5 | 0.80 | 6.0 | 0.84 | 3.9 | 0.86 | 3.6 |
| 4 to 6 | 717 | 0.68 | 31.7 | 0.68 | 19.7 | 0.70 | 9.7 | 0.71 | 5.3 | 0.64 | 4.4 |
| 7 to 10 | 334 | 0.63 | 41.0 | 0.63 | 24.9 | 0.60 | 11.6 | 0.22 | 5.7 | 0.00 | 4.5 |
| 11 to 100 | 622 | 0.55 | 57.8 | 0.51 | 34.1 | 0.26 | 14.7 | 0.00 | 6.4 | 0.00 | 4.6 |
| 101 to 1000 | 81 | 0.23 | 96.7 | 0.00 | 51.6 | 0.00 | 19.1 | 0.00 | 7.1 | 0.00 | 4.7 |

Table 7: **Coverage and size conditional on difficulty.** We report coverage and size of RAPS sets using ResNet-152 with $k_{reg} = 5$ and varying $\lambda$ (recall that $\lambda = 0$ is the APS procedure). The desired coverage level is 90%. The 'difficulty' is the ranking of the true class's estimated probability.

| | Violation at $\alpha = 10\%$ | | Violation at $\alpha = 5\%$ | |
|---|---|---|---|---|
| Model | APS | RAPS | APS | RAPS |
| ResNeXt101 | 0.090 | **0.049** | 0.048 | **0.021** |
| ResNet152 | 0.069 | **0.038** | 0.037 | **0.017** |
| ResNet101 | 0.073 | **0.041** | 0.038 | **0.017** |
| ResNet50 | 0.069 | **0.037** | 0.037 | **0.016** |
| ResNet18 | 0.046 | **0.025** | 0.032 | **0.019** |
| DenseNet161 | 0.080 | **0.047** | 0.040 | **0.016** |
| VGG16 | 0.046 | **0.022** | 0.030 | **0.022** |
| Inception | 0.085 | **0.045** | 0.043 | **0.023** |
| ShuffleNet | 0.061 | **0.033** | 0.035 | **0.020** |

Table 8: **Adaptiveness results after automatically tuning $\lambda$.** We report the median size-stratified coverage violations of APS and RAPS over 10 trials. See Appendix E.2 for experimental details.

# E  Choosing $k_{reg}$ and $\lambda$ to optimize set size and adaptiveness

This section describes two procedures for picking $k_{reg}$ and $\lambda$ that optimize for set size or adaptiveness, outperforming APS in both cases.

## E.1  Optimizing set size with RAPS

---
**Algorithm 4** Adaptive Fixed-K

**Input:** $\alpha$; $I \in \{1, ..., K\}^{n \times K}$, and one-hot $y \in \{0, 1\}^K$ corresponding respectively to the classes from highest to lowest estimated probability mass, and labels for each of $n$ examples in the dataset
1: **procedure** GET-KSTAR($\alpha$,I,y)
2:     **for** $i \in \{1, \cdots, n\}$ **do**
3:         $L_i \leftarrow \{ j : I_{i,j} = y_i \}$
4:     $\hat{k}^* \leftarrow$ the $\lceil (1 - \alpha)(1 + n) \rceil$ largest value in $\{L_i\}_{i=1}^n$
5:     **return** $\hat{k}^*$
**Output:** The estimate of the smallest fixed size set that achieves coverage, $\hat{k}^*$

---

To produce Tables 1, 5, 2, and 6, we chose $k_{reg}$ and $\lambda$ adaptively. This required an extra data splitting step, where a small amount of *tuning data* $\{x_i, y_i\}_{i=1}^m$ were used to estimate $k^*$, and then $k_{reg}$ is set to $\hat{k}^*$. Taking $m \approx 1000$ was sufficient, since the algorithm is fairly insensitive to $k_{reg}$ (see Table 3). Then, $\hat{k}^*$ was calculated with Algorithm 4. We produced the Imagenet V2 tables with $m = 1000$ and the Imagenet tables with $m = 10000$.

After choosing $\hat{k}^*$, we chose $\lambda$ to have small set size. We used the same tuning data to pick $\hat{k}^*$ and $\lambda$ for simplicity (this does not invalidate our coverage guarantee since conformal calibration still uses fresh data). A coarse grid search on $\lambda$ sufficed, since small parameter variations have little impact on RAPS. For example, we chose the $\lambda \in \{0.001, 0.01, 0.1, 0.2, 0.5\}$ that achieved the smallest size on the $m$ holdout samples in order to produce Tables 1, 5, 2, and 6. We include a subroutine that automatically chooses $\hat{k}^*$ and $\lambda$ to optimize size in our GitHub codebase.

### E.2 OPTIMIZING ADAPTIVENESS WITH RAPS

In this appendix, we show empirically that RAPS with an automatically chosen set of $k_{reg}$ and $\lambda$ improves the adaptiveness of APS. Recall our discussion in Section 4 and Proposition 3, wherein we propose size-stratified coverage as a useful definition of adaptiveness in image classification. After picking $k_{reg}$ as in Appendix E, we can choose $\lambda$ using the same tuning data to optimize this notion of adaptiveness.

We now describe a particular manifestation of our adaptiveness criterion that we will use to optimize $\lambda$. Consider disjoint set-size strata $\{S_i\}_{i=1}^{i=s}$, where $\bigcup_{j=1}^{j=s} S_i = \{1, \ldots, |\mathcal{Y}|\}$. Then define the indexes of examples stratified by the prediction set size of each example from algorithm $\mathcal{C}$ as $\mathcal{J}_j = \{i : |\mathcal{C}(X_i, Y_i, U_i)| \in S_j\}$. Then we can define the *size-stratified coverage violation* of an algorithm $\mathcal{C}$ on strata $\{S\}_{i=1}^{i=s}$ as

$$\text{SSCV}\big(\mathcal{C}, \{S\}_{j=1}^{j=s}\big) = \sup_j \left| \frac{|\{i : Y_i \in \mathcal{C}(X_i, Y_i, U_i), i \in \mathcal{J}_j\}|}{|\mathcal{J}_j|} - (1-\alpha) \right|. \tag{5}$$

In words, Eq. (5) is the worst-case deviation of $\mathcal{C}$ from exact coverage when it outputs sets of a certain size. Computing the size-stratified coverage violation thus only requires post-stratifying the results of $\mathcal{C}$ on a set of labeled examples. If conditional coverage held, the worst stratum coverage violation would be 0 by Proposition 3.

To maximize adaptiveness, we'd like to choose $\lambda$ to minimize the size-stratified coverage violation of RAPS. Write $\mathcal{C}_\lambda$ to mean the RAPS procedure for a fixed choice of $k_{reg}$ and $\lambda$. Then we would like to pick

$$\lambda = \arg\min_{\lambda'} \text{SSCV}(\mathcal{C}_{\lambda'}, \{S\}_{j=1}^{j=s}). \tag{6}$$

In our experiments, we choose a relatively coarse partitioning of the possible set sizes: 0-1, 2-3, 4-10, 11-100, and 101-1000. Then, we chose the $\lambda \in \{0.00001, 0.0001, 0.0008, 0.001, 0.0015, 0.002\}$ which minimized the size-stratified coverage violation on the tuning set. The results in Table 8 show RAPS always outperforms the adaptiveness of APS on the test set, even with this coarse, automated choice of parameters. The table reports the median size-stratified coverage violation over 10 independent trials of APS and RAPS with automated parameter tuning.

## F COMPARISON WITH LEAST AMBIGUOUS SET-VALUED CLASSIFIERS

In this section, we compare RAPS to the Least Ambiguous Set-valued Classifier (LAC) method introduced in Sadinle et al. (2019), an alternative conformal procedure that is designed to have small sets. The LAC method provable gives the smallest possible average set size in the case where the input probabilities are correct, with the idea that these sets should be small even when the estimated probabilities are only approximately correct. In the notation of this paper, the LAC method considers nested sets of the following form:

$$\mathcal{C}^{\text{LAC}}(x, \tau) := \{y : \hat{\pi}_x(y) \geq 1 - \tau\},$$

which can be calibrated using as before in using $\hat{\tau}_{\text{ccal}}$ from Eq. (3).

We first compare naive, APS, RAPS, and LAC in terms of power and coverage in Table 9. In this experiment, we tuned RAPS to have small set size as described in Appendix E.1. We see that LAC also achieves correct coverage, as expected since it is a conformal method and satisfies the guarantee from Theorem 1. We further see that it has systematically smaller sets that RAPS, although the difference is slight compared to the gap between APS and RAPS or APS and LAC.

We next compare RAPS to LAC in terms of adaptiveness, tuning RAPS as in Section E.2. First, in Table 10, we report on the coverage of LAC for images of different difficulties, and see that LAC has dramatically worse coverage for hard images than for easy ones. Comparing this to RAPS in Table 7, we see that RAPS also has worse coverage for more difficult images, although the gap is much smaller for RAPS. Next, in Table 11, we report on the SSCV metric for of adaptiveness (and conditional coverage) for APS, RAPS, and LAC. We find that APS and RAPS have much better adaptiveness than LAC, with RAPS being the overall winner. The results of all of these comparisons are expected: LAC is not targeting adpativeness and instead trying to achieve the smallest possible set size. It succeeds at its goal, sacrificing adaptiveness to do so.

| | Accuracy | | Coverage | | | | | Size | | | | |
|---|---|---|---|---|---|---|---|---|---|---|---|---|
| Model | Top-1 | Top-5 | Top K | Naive | APS | RAPS | LAC | Top K | Naive | APS | RAPS | LAC |
| ResNeXt101 | 0.793 | 0.945 | 0.900 | 0.889 | 0.900 | 0.900 | 0.900 | 2.42 | 17.2 | 19.9 | 2.01 | 1.65 |
| ResNet152 | 0.783 | 0.941 | 0.900 | 0.894 | 0.900 | 0.900 | 0.900 | 2.64 | 9.68 | 10.4 | 2.09 | 1.76 |
| ResNet101 | 0.774 | 0.936 | 0.900 | 0.895 | 0.900 | 0.900 | 0.900 | 2.83 | 10.0 | 10.8 | 2.25 | 1.87 |
| ResNet50 | 0.761 | 0.929 | 0.899 | 0.896 | 0.900 | 0.900 | 0.900 | 3.13 | 11.7 | 12.3 | 2.55 | 2.05 |
| ResNet18 | 0.698 | 0.891 | 0.900 | 0.895 | 0.900 | 0.900 | 0.900 | 5.74 | 15.3 | 16.1 | 4.38 | 3.64 |
| DenseNet161 | 0.771 | 0.936 | 0.900 | 0.894 | 0.900 | 0.900 | 0.900 | 2.84 | 11.2 | 12.0 | 2.29 | 1.90 |
| VGG16 | 0.716 | 0.904 | 0.900 | 0.896 | 0.901 | 0.900 | 0.900 | 4.75 | 13.4 | 14.1 | 3.54 | 3.00 |
| Inception | 0.695 | 0.886 | 0.899 | 0.884 | 0.900 | 0.899 | 0.900 | 6.27 | 74.8 | 88.8 | 5.24 | 4.06 |
| ShuffleNet | 0.694 | 0.883 | 0.900 | 0.892 | 0.900 | 0.899 | 0.900 | 6.45 | 28.8 | 32.1 | 5.01 | 4.14 |

Table 9: **Results on Imagenet-Val.** We report coverage and size of the optimal, randomized fixed sets, `naive`, `APS`, `RAPS` , and the LAC sets for nine different Imagenet classifiers. The median-of-means for each column is reported over 100 different trials at the 10% level. See Section 3.1 for full details.

| difficulty | count | cvg | sz |
|---|---|---|---|
| 1 | 15668 | 1.00 | 1.5 |
| 2 to 3 | 2578 | 0.81 | 2.6 |
| 4 to 6 | 717 | 0.23 | 3.0 |
| 7 to 10 | 334 | 0.00 | 2.9 |
| 11 to 100 | 622 | 0.00 | 2.7 |
| 101 to 1000 | 81 | 0.00 | 2.4 |

Table 10: **Coverage and size conditional on difficulty.** We report coverage and size of the LAC sets for ResNet-152.

| Model | Violation at $\alpha = 10\%$ | | | Violation at $\alpha = 5\%$ | | |
|---|---|---|---|---|---|---|
| | APS | RAPS | LAC | APS | RAPS | LAC |
| ResNeXt101 | 0.086 | **0.047** | 0.217 | 0.047 | **0.022** | 0.127 |
| ResNet152 | 0.067 | **0.039** | 0.156 | 0.04 | **0.021** | 0.141 |
| ResNet101 | 0.075 | **0.060** | 0.152 | 0.039 | **0.015** | 0.120 |
| ResNet50 | 0.071 | **0.042** | 0.131 | 0.037 | **0.014** | 0.109 |
| ResNet18 | 0.050 | **0.024** | 0.196 | 0.031 | **0.021** | 0.059 |
| DenseNet161 | 0.076 | **0.055** | 0.140 | 0.039 | **0.016** | 0.110 |
| VGG16 | 0.051 | **0.023** | 0.165 | 0.029 | **0.019** | 0.070 |
| Inception | 0.086 | **0.042** | 0.181 | 0.043 | **0.023** | 0.135 |
| ShuffleNet | 0.058 | **0.030** | 0.192 | 0.033 | **0.019** | 0.045 |

Table 11: **Adaptiveness results after automatically tuning $\lambda$.** We report the median SSCV of `APS` `RAPS` and LAC over 10 trials. See Appendix E.2 for experimental details.

