# OpenReview forum: "Uncertainty Sets for Image Classifiers using Conformal Prediction"
_ICLR.cc/2021/Conference — ICLR 2021 Spotlight_

### Official Review · AnonReviewer1 · 2020-10-27
**elegant regularization strategy in constructing prediction sets; could use more discussion on issues related to high-stakes decisions**

**Rating:** 7
**Confidence:** 4

**Review:**

This paper proposes a regularized generalization of adaptive prediction sets (APS by Romano et al 2020) that results in smaller prediction sets that still maintain the correct coverage level for statistical validity.

The basic idea of the new regularization is very well-explained and quite elegant: roughly speaking, there is an indifference between classes of low probability so we should penalize including them in the prediction set. Despite the simplicity of this idea, I consider this paper's contribution to be a significant advance and I suspect this idea to be useful beyond classification problems with images, although from my understanding, a crucial assumption is that the number of classes needs to be sufficiently large (e.g., the regularization will not really yield any benefit if it's binary classification). I also found the experiments to be well-chosen.

My main comment instead is perhaps more of a follow-up: would it be possible to get a conditional guarantee rather than a marginal guarantee (Proposition 1) by adapting proof ideas from Tibshirani et al's "Conformal Prediction Under Covariate Shift" (2019)? This seems more aligned with the healthcare related question the paper starts the paper with. In the healthcare context, the patient would ideally want the coverage to be conditional rather than marginal (i.e., in Proposition 1, for the probability that's being sandwiched, we want a version conditioned on $X_{n+1}$ landing near an anchor feature vector to suggest randomly sampling feature vectors *similar* to, for instance, a specific patient in the healthcare context). Separately, it would be interesting if the approach from Tibshirani et al can help RAPS construct even smaller prediction sets than what you get in bold for Table 2 (in the marginal coverage setting and not worrying about conditional coverage).

Strengths:
- elegant, well-explained method with crisp theory
- nice suite of experiments
- helps popularize conformal prediction, which I think more machine learning researchers should know about

Weakness:
- the paper opens up with a high-stakes motivation but I don't think it really gets back to this high-stakes problem properly; I'd suggest discussing conditional coverage more (also some high-stakes classification problems have very few classes, which the proposed regularization doesn't provide much benefit for?)

---

> ### Author Response · Authors · 2020-11-25
> **Brief response and thanks; see supplementary material for summary of major revision and point-by-point responses**
>
> We thank this reviewer for their favorable comments and helpful suggestions.  We enjoyed reading the Tibshirani et al. paper and welcomed the opportunity to engage the reader in a detailed discussion of conditional coverage and adaptiveness, partially motivated by the local notion introduced in that paper. Our new manuscript now provides, among many new changes, a new Section 4 describing a straightforward notion of adaptiveness appropriate for image classification and an automated choice of RAPS parameters to directly optimize that criterion. Furthermore, a different automated choice of parameters optimizes size directly, not worrying about conditional coverage, improving our results and making the procedure easier to use for computer vision practitioners.  The reviewer's comments thus produced quite a positive impact on our work; we are grateful.
>
> We responded to the reviewers' criticisms with a major revision of the paper and detailed point-for-point changes in the text. In order to organize this response more effectively, we refer the reviewer to our response document, which we have added in the "Supplementary Materials" section of our revised submission. The first two pages are a summary intended for all reviewers to read. Then we include point-by-point responses to each reviewer, and at the end of the document, we include a marked-up version of our submission, where blue text is new text.
>
> **Please download the supplementary material .zip file above to see the detailed responses to the reviewer's comments.**

---

### Official Review · AnonReviewer2 · 2020-10-27
**A paper with some interesting insights, but also some significant weaknesses**

**Rating:** 7
**Confidence:** 4

**Review:**

## Summary
The paper proposes a new conformalized procedure for computing uncertainty sets in classification tasks. The key feature of the method is that the size of the uncertainty sets are regularized via a penalty on the size. The issue of large uncertainty sets produced by conformalized procedures is an interesting one, which the paper does well to highlight. The proposed solution of using an additive regularizer is reasonable, and appears to be effective for sensible choices of the hyper-parameters. However, the paper has some significant weaknesses.

---
## Strengths
1. The method is easy to understand and to implement.
2. The method satisfies some meaningful theoretical guarantees. (The proofs have been checked for correctness.)

## Weaknesses (and Questions)
1. I am not 100% convinced that one would _want_ to prune extremely large uncertainty sets arising out of the APS procedure. Because the marginal coverage must be maintained, this forces some small uncertainty sets to be enlarged. However, in many practical classification tasks (including the experiments in this paper), it tends also to be the case that difficult examples are less common than the easier ones. It is hard for me to imagine that one would ever be in a situation that one would wish to over-cover typical examples and undercover atypical ones to an even larger extent, even if the over-coverage is only by one or two classes.

However, perhaps there are real world situations in which this is precisely the case. It would be nice if the authors could offer additional insights on this point.

2. The method requires the user to choose two hyper-parameters $k_{reg}$ and $\lambda$, with some combinations leading to decidedly less desirable results. For the particular case of $k_{reg} = k_*$, perhaps one could look at the RAPS solution as "interpolating" between the APS solution and the top-$k_*$ solution. Related to #1 above, it isn't clear to me why this would be desirable in any way. Furthermore, although the hyper-parameters can be loosely interpreted as having to do with the size and the tradeoff with adaptivity, respectively, the precise relationship does not appear to be understood for general cases, making it more challenging to predict the behavior of the method. This is in contrast to the APS, which can be motivated via an oracle procedure with certain optimality properties.

---
## Recommendation
A borderline reject. The paper has some significant weaknesses, but does contain some interesting insights.

---
## Additional Feedback
1. Section B in the appendix is an essential reading for anyone looking to implement the procedure. I think it ought to be included in the paper.
2. It looks to me that the method could suffer from poor choices of $k_{reg}$ and $\lambda$. To prevent such choices, if $k_{reg} = k_*$ is in some sense an oracle choice, wouldn't it be better to give the procedure _with_ the particular hyper-parameter tuning procedure already incorporated?
3. Personally, I found Tables 4 and 5 far more informative than Figure 4. I would prefer to see the tables in the paper.
4. I am not sure if the median-of-means is the right metric for performance comparisons. One way to view the RAPS is that it "tames" the tail of the distribution of the size of uncertainty sets produced by the APS. Personally, I would prefer a direct comparison of the distributions, as in the tables in the appendix. However, even if a summary measure must be used, there are probably better options.
5. I find the comment in Section 3.2 "The result shows that our method can still provide exact coverage under a significant distribution shift" somewhat misleading. In a split conformal setup, the only distribution shift of any import (with regards to the marginal coverage guarantee) is the one between the calibration set and a new test point.

## Update
Please see my reply to the authors' message.

---

> ### Author Response · Authors · 2020-11-25
> **Brief response and thanks; see supplementary material for summary of major revision and point-by-point responses**
>
> We thank this reviewer for their comments, which pointed out the two major flaws in our initial submission: the lack of justification for pruning extremely large uncertainty sets, and the choices of k_{reg} and \lambda.  Inspired by these comments and similar concerns from other reviewers, we added new sections, experiments, propositions, and tables to address these points.  In particular, we find that the APS procedure systematically overcovers hard examples and undercovers easy ones, a strange behavior that RAPS fixes with a small choice of \lambda.  We also describe a subroutine that automatically chooses \lambda to maximize the adaptiveness of RAPS, and show that it outperforms APS on that front as well. In other words, RAPS does not have to trade off adaptiveness for size, although we do include another subroutine to automatically optimize RAPS to have small size. These improvements and many others would not have been made without the reviewer's helpful criticism, and we are grateful.
>
> We responded to the reviewers' criticisms with a major revision of the paper and detailed point-for-point changes in the text. In order to organize this response more effectively, we refer the reviewer to our response document, which we have added in the "Supplementary Materials" section of our revised submission. The first two pages are a summary intended for all reviewers to read. Then we include point-by-point responses to each reviewer, and at the end of the document, we include a marked-up version of our submission, where blue text is new text.
>
> **Please download the supplementary material .zip file above to see the detailed responses to the reviewer's comments.**

---

### Official Review · AnonReviewer3 · 2020-10-28
**Reasonable, but limited, extension to create smaller conformal prediction sets**

**Rating:** 7
**Confidence:** 4

**Review:**

=== Summary ===

In this paper, the authors propose a regularized conformal score for use in a conformal prediction framework. This regularizer is motivated by the instabilities of top-p variations on conformal scores (cf. Romano et. al., 2020) and the resulting high-variance in output conformal prediction set sizes. The proposed regularizer smooths top-p scores with top-k scores, which empirically results in more robust predictive sets. The authors also perform a large-scale evaluation on ImageNet with modern architectures, which serves as a helpful benchmark for conformal prediction algorithms.

For those not familiar with the goal of conformal prediction: it is a method to provide prediction *sets* in-place of single (top-1) predictions, where the sets satisfy marginal probabilistic performance guarantees (i.e., the correct answer is included in the set with probability ≥ 1 - alpha).

=== Justification for Score ===

The main contribution of the paper (RAPS) is reasonable, but has perhaps somewhat limited novelty when taken in context with the broader conformal prediction literature. The paper, at least at its current version, also has several presentational and clarity issues which makes it somewhat hard to follow at points, and some of the results difficult to verify. I am willing to be convinced to increase my score if at least 1) the language of the introduction is reworked to reflect the broad conformal prediction literature, and 2) it is demonstrated that RAPS significantly improves over a naive "adaptive"  (i.e., conformalized) top-k baseline (see "Concerns").

=== Strengths ===

+ Uncertainty quantification is a relevant and timely topic.

+ Conformal prediction and the resulting uncertainty sets they produce is also a practical and important instantiation of uncertainty quantification/prediction with performance guarantees.

+ The proposals put forth in the paper are interesting and potentially useful.

=== Concerns ===

- Overall, there are unfortunately several clarity and presentation issues (see also "Minor Comments" below for more on this). At a high level, however, it seems to me that the framing in this paper is confusing when it comes to distinguishing the contributions put forth here, versus those of the broader conformal prediction literature. This stems from language in the abstract, such as "we present an algorithm that..." or in the introduction, such as "Our paper describes a novel method for constructing prediction sets...". Yet, the main contribution appears to rather be a particular formulation of a *conformal scoring function*, i.e., S(x, y) := \rho_x(y) + \hat{\pi_x}(y) * u + \lambda * (o_x(y) - k_reg)^+.  Similar variants (as the authors do point out) have already been proposed in Romano et. al., 2020 and Cauchois et. al., 2020. I would strongly recommend updating the language and introductory claims so as not to mislead readers.

- It should be clarified that the guarantee expressed in Eq. 1 is *marginal*, i.e., taken over all permutations of (X_1, Y_1), ..., (X_{n+1}, Y_{n+1}). It also should hold for exchangeable rather than just i.i.d. samples; I didn't see a reason in the method to weaken this standard CP result. Finally, the comment on "require only 1000 held-out data points in practice" is confusing to me... in practice to achieve what? The guarantee in Eq. 1 should hold for all finite samples.

- Eq. 1 stipulates a ≥ 1 - alpha criterion (which the methods support)---but Section 2 begins by saying "exactly 1 - alpha" coverage. Can you clarify?

- In the introduction it states, "for examples where the model is not confident, naive must select many classes...". It's not immediately apparent why this is a problem. If the model probability p(y | x) is indeed calibrated, then it seems that naive should indeed be the oracle method (in order to guarantee both marginal *and* conditional coverage). Of course, softmax outputs in standard deep nets can be far from calibrated, which leads to the suboptimal behavior. But this doesn't seem to be the problem the introduction is pin-pointing.

- In fact, related to the above, both Romano et. al., 2020 and Cauchois et. al., 2020 motivate their conformalizations as an approach to asymptotically get conditional coverage (assuming p(y | x) converges with more data and better models). Accordingly they also measure a "worst slab coverage" to quantify how far the method deviates from conditional coverage in the worst case. The proposed top-k regularizer breaks this asymptotic property, which at least should be something to note relative to those related works. It would be even better if some additional experiments in the appendix showed what realized trade-off, if any, there is between WSC and tighter set sizes due to regularization.

- Proposition 2 claims strictly better than top-k performance, but the proof seems to suggest that this is rather only \subseteq?

- In a similar vein, Proposition 2 makes comparisons to top-k performance, but (adaptive) top-k isn't shown in Table 1. Only top-1 and top-5 are. If simply using the rank as the conformal score, how does the set size compare to RAPS? Note that top-k will have many ties, so it also makes sense to use a randomized variant of top-k (i.e., S(x, y) := o_x(y) + u) for fair comparison.

- lambda and k_reg are hyper-parameters that must be set. In addition to the n calibration data points, it would seem that one would also require additional development data to use to pick these. It's also unclear what data was used to do the platt scaling. If on the same calibration data, it's not clear to me that this is retains exchangeability, which will affect the formal guarantees.

=== Minor Comments ===

* The pseudo code in the algorithms, which uses a mixture of python-style notation and mathematical indexing (e.g., I_i[j] in Algorithm 2), is quite hard to follow. I suggest sticking to one.

* Algorithm 1 line 7 is missing a sum of some sort around scores[0: L].

* Algorithm 2 lines 3-4 has sloppy notation: it appears a set (L_i) is being added to a scalar (k_reg + 1).

* Though I certainly appreciate the page constraints, Tables 1 & 2 are too small to easily read. I don't think that the Top-1 and Top-5 accuracy add much (see earlier comments on adding a conformalized top-k result).

* Similarly, the left side of Fig. 4 is hard to read.

=== Response After Rebuttal ===

Thank you for what I found to be an impressive and convincing rebuttal, both in terms of the detailed responses to every comment---and not to mention the extensive rewrite of this paper in the limited amount of time available. In short, I think that the paper is significantly improved, and is now at a level of quality that I feel warrants acceptance. As such, I am choosing to raise my initial score of 4 to 7 (good paper, accept).

As the revised version introduces many changes, I have some new (minor) comments:

- The last line of the abstract claims "sets that are often factors of 5 to 10 smaller". This is true for APS and NAIVE, but not adaptive top-k. You might wish to qualify this statement as to which comparison you are referring to.

- Thank you for clarifying the introduction w.r.t. problems with NAIVE. The third "problem" seems a bit funny to me; it doesn't identify a *problem* with NAIVE, but rather that other methods are better (which seems to stem from problems 1 and 2). Can you clarify this point to make it more precise?

- Typo, related work: "[...] data-splitting version known as *split conformal prediction* THAT enables conformal prediction [...]"

- Related work when talking about equal coverage per class, you might want to cite the earlier reference of mondrian conformal prediction: Vladimir Vovk, David Lindsay, Ilia Nouretdinov, and Alex Gammerman. Mondrian confidence machine. Technical Report, 2003.

- Theorem 1's remark on "this result is not new". Not to nitpick, but I'm not sure which aspects of Thm. 1 makes it "first appear" in Papadopoulus et. al. and Hechtlinger et. al? This result can likely be found in (or if not explicitly in, then derived from) from late 1990's work by Vovk and co. on Ridge Regression Confidence Machines, Transductive Confidence Machines, etc. Or the journal/book papers (i.e., Algorithmic Learning in a Random World, Hedging Predictions in Machine Learning, etc).

(It goes without saying, however, that this revision is much better in terms of proper citation than before!)

- Proposition 2 seems to have some funny crowding around the \subseteq operator.

-  Nice work on adding section 4. In the proof of proposition 3 I believe you accidentally left out set size (| |) around C(X) on the r.h.s.

- In the discussion, you mention improving efficiency via prediction cascades. You might be interested to refer to a concurrent ICLR submission (https://openreview.net/forum?id=tnSo6VRLmT) which does precisely this, and to which your work on larger label spaces seems quite complementary.

---

> ### Author Response · Authors · 2020-11-25
> **Brief response and thanks; see supplementary material for summary of major revision and point-by-point responses**
>
> We thank this reviewer for their very helpful and detailed constructive criticism.  At the reviewer's suggestion, we have tried to make  it abundantly clear that our method is a modification of an already existing conformal method early on in the paper. Furthermore, we thought the inclusion of an adaptive top-k baseline was a great idea; our new tables include this baseline, and RAPS consistently outperforms it. We thank the reviewer for providing these clear and precise guidelines for the score increase and believe we have met them.  Furthermore, the reviewer's suggestion of using worst-slab coverage as a proxy for adaptiveness via conditional coverage were inspiring to us; we welcomed the opportunity to add a new section of our paper wherein we develop a notion of adaptiveness more appropriate for image classification. Using this metric, we can optimize RAPS to get closer to conditional coverage than APS with an automatic choice of k_{reg} and \lambda.  These comments made the paper much better and we are immensely grateful.
>
> We responded to the reviewers' criticisms with a major revision of the paper and detailed point-for-point changes in the text. In order to organize this response more effectively, we refer the reviewer to our response document, which we have added in the "Supplementary Materials" section of our revised submission. The first two pages are a summary intended for all reviewers to read. Then we include point-by-point responses to each reviewer, and at the end of the document, we include a marked-up version of our submission, where blue text is new text.
>
> **Please download the supplementary material .zip file above to see the detailed responses to the reviewer's comments.**

---

### Official Review · AnonReviewer4 · 2020-10-29

**Rating:** 7
**Confidence:** 4

**Review:**

##########################################################################

Summary:

 Prediction sets are used to quantify the uncertainty of classification. The naive approach which include the labels until a pre-specified coverage probability is satisfied often leads to large prediction sets. Adaptive Prediction Sets (APS) can output prediction sets with desired coverage but set sizes are still not satisfyingly small and the results are unstable, especially when many probability estimations fall into the tail of the distribution.

In order to make the prediction stable and sets as narrow as possible under pre-specified coverage probability, this paper extends APS to Regularized Adaptive Prediction Sets (RAPS) by penalizing those class with small probabilities beyond k many classes already included, which leads to a small prediction size. The regularization is an interesting idea in terms of minimizing prediction sets, which is different from previous works where most of them directly minimize a quantity related to the cardinality of prediction sets or intervals. Empirically, compared with other set-valued classifiers extracting information from the same base model CNNs, the proposed method outperforms significantly in terms of set sizes when fixing pre-specified coverage. Moreover, this work shows adaptiveness: it tries to allow large prediction size for difficult instances and small prediction size for easy instances.

##########################################################################

Reasons for score:

Overall, I vote for accepting. I think the method is well motivated and the solution is simple and portable (can be applied to many base methods).  However, there could be more discussions on several aspects of the problem.


##########################################################################

Pros:

1. Studies an important problem.

2. The proposed method is easy to implement and can be applied to general scores or be used to improve base conformal prediction methods.

3. Very impressive empirical performance.

 ##########################################################################

Cons:

1. Theoretically the "optimal" set-valued classifier is based on P(Y=k | X=x). In this sense, the naive approach can be viewed as a plug-in approach when the score is an estimate of P(Y=k | X=x). When regularization is applied, something must be lost. This is as much like in lasso for high-dimensional regression, a penalty function makes the coefficient estimate biased (to trade for sparisity). It is unclear what is lost here with regularization. Is the solution no longer "Fisher consistent" in a sense?

2. More to the point: it seems that the proposed method is cut out for problems in which there are MANY classes. I wonder whether it will perform just as well for traditional problems in which there are only a few classes (like in the medical field.)

3. Choose a good value for k_reg and lambda seems to be critical. How sensitive is the result to k_reg? Is there any general theory or guidelines about tuning parameter lambda? In the experiments, the validation (calibration) data sets have huge sample size, which may be common in image data domains, but can be unrealistic for broader applications domains. I wonder if the good performance is largely relying on the large validation (calibration) sample size.

##########################################################################

Questions during rebuttal period:

The goal of narrowing prediction set size is achieved with the help of regularization, which does not directly try to minimize the cardinality of the prediction set. Can we theoretically prove it is asymptotically optimal? Any comparison to these direct approaches? There is a literature called high-quality prediction interval which directly minimizes the prediction size.

+ Tim Pearce, Mohamed Zaki, Alexandra Brintrup, and Andy Neely. High-quality prediction intervals for deep learning: A distribution-free, ensembled approach. arXiv preprint arXiv:1802.07167, 2018.

Any comments on the relation of the uncertainty set approach with the classification with rejection/abstain methods?

+ Zhang, C., Wang, W. and Qiao, X. (2018), “On Reject and Refine Options in Multicategory Classification,” Journal of the American Statistical Association, 113 (522), pp. 730–745.
+ Ramaswamy HG, Tewari A, Agarwal S. Consistent algorithms for multiclass classification with an abstain option. Electronic Journal of Statistics. 2018;12(1):530-54.

#########################################################################

Small comments:

Most of the figures and tables are far away from the descriptions which makes it hard to read.

---

> ### Author Response · Authors · 2020-11-25
> **Brief response and thanks; see supplementary material for summary of major revision and point-by-point responses**
>
> We thank the reviewer for their comments, particularly their suggestions about APS approximating conditional coverage P(Y=k | X=x) and also choosing k_{reg} and \lambda.  These fundamental issues deserved quite a bit more development in the paper, so we have included new sections, experiments, propositions, and tables to address them.
>
> We responded to the reviewers' criticisms with a major revision of the paper and detailed point-for-point changes in the text. In order to organize this response more effectively, we refer the reviewer to our response document, which we have added in the "Supplementary Materials" section of our revised submission. The first two pages are a summary intended for all reviewers to read. Then we include point-by-point responses to each reviewer, and at the end of the document, we include a marked-up version of our submission, where blue text is new text.
>
> **Please download the supplementary material .zip file above to see the detailed responses to the reviewer's comments.**

---

### Author Response · Authors · 2020-11-25
**Response to Reviewers**

We responded to the reviewers' criticisms with a major revision of the paper and detailed point-for-point changes in the text. In order to organize this response more effectively, we refer the reviewer to our response document, which we have added in the "Supplementary Materials" section of our revised submission.  The first two pages are a summary intended for all reviewers to read.  Then we include point-by-point responses to each reviewer, and at the end of the document, we include a marked-up version of our submission, where blue text is new text.

**Please download the supplementary material .zip file above to see the detailed responses to the reviewers.**

---

### Decision · Program_Chairs · 2021-01-07
**Final Decision**

**Decision:**

Accept (Spotlight)

**Comment:**

The reviewers all agreed that the paper is a solid contribution.

Pros:
- A simple and reasonable extension to adaptive prediction sets that performs well empirically.
- The procedure presented is versatile (i.e. can be applied to general scores or be used to improve base conformal prediction methods).
- A very thorough experimental analysis, including large datasets (i.e. Imagenet) and a wide range of model architectures including ResNet-152.
- Some formal theoretical guarantees are provided for the procedure, although they appear to be straightforward.

Cons:
- Limited technical novelty.

Overall, I recommend a spotlight because the reviewers felt that the topic of predictive uncertainty is of interest to the broader ML and computer vision community, and the paper can have a potentially large impact in popularizing conformal methods as a viable uncertainty estimation method.